# Addressing Combative Behaviour in Spanish Bulls by Measuring Hormonal Indicators

**DOI:** 10.3390/vetsci11040182

**Published:** 2024-04-22

**Authors:** Juan Carlos Illera, Francisco Jimenez-Blanco, Luis Centenera, Fernando Gil-Cabrera, Belen Crespo, Paula Rocio Lopez, Gema Silvan, Sara Caceres

**Affiliations:** Department Animal Physiology, Veterinary Medicine School, Complutense University of Madrid (UCM), 28040 Madrid, Spain; jcillera@ucm.es (J.C.I.); jmesa58@gmail.com (F.J.-B.); luiscentenera@hotmail.com (L.C.); fernando.gil@congreso.es (F.G.-C.); belencre@ucm.es (B.C.); paularlo@ucm.es (P.R.L.); sacacere@ucm.es (S.C.)

**Keywords:** fighting bull, serotonin, dopamine, testosterone

## Abstract

**Simple Summary:**

Aggressiveness in fighting bulls is a natural characteristic of this breed, but little is known regarding it physiological mechanisms. Hormones such as serotonin, dopamine and testosterone are known to be involved in the development of aggressive behaviour. This study determines that serotonin and dopamine levels are correlated with combative behaviour, and its determination in calves is a useful indicator for the selection of combative bulls.

**Abstract:**

The fighting bull is characterised by its natural aggressiveness, but the physiological mechanisms that underlie its aggressive behaviour are poorly studied. This study determines the hormonal component of aggressiveness in fighting bulls by analysing their behaviour during a fight and correlating it to their serotonin, dopamine and testosterone levels. We also determine whether aggressive behaviour can be estimated in calves. Using 195 animals, samples were obtained when the animals were calves and after 5 years. Aggressiveness scores were obtained by an observational method during bullfights, and serotonin, dopamine and testosterone levels were determined in all animals using validated enzyme immunoassay kits. The results revealed a strong correlation of serotonin and dopamine levels with aggressiveness scores in bulls during fights, but no correlation was found with respect to testosterone. These correlations led to established cut-off point and linear regression curves to obtain expected aggressiveness scores for calves at shoeing. There were no significant differences between the expected scores obtained in calves and the observed scores in bulls. Therefore, this study demonstrates that hormone determination in calves may be a great indicator of combativeness in bulls and can reliably be used in the selection of fighting bulls.

## 1. Introduction

Aggressiveness is defined as a tendency to act or respond violently. It is considered a primitive and conserved animal behaviour. The domestication of different animal species, including cattle, has led to the selection of more docile animals, discarding animals with aggressive traits [1]. However, in some species, aggressiveness is a trait used for the selection of certain breeds, such as in the Lidia cattle breed. Popularly known as the fighting bull, this cattle breed is characterised by a strong physical appearance, extraordinary strength and natural aggressiveness [2]. Lidia cattle are mainly used for breeding under extensive conditions in central and southern Spain, France, Portugal and Latin American countries. This breed, also known as “toro bravo”, refers to the male specimens of a heterogeneous bovine population developed, selected and bred for use in different bullfighting spectacles. They come from the indigenous breeds of the Iberian Peninsula. They are characterised by atavistic instincts of defence and temperament, which are characterised by so-called “bravery”, as well as physical attributes such as large forward-facing horns and powerful locomotor apparatus [2,3].

Its breeding system is extensive, and its importance in traditional shows has promoted wide diversity in the breed given the requirement for bulls with different behaviours and characteristics that have been traditionally selected for aggressiveness, ferocity and mobility. The isolation of individuals from the herd, their confinement or their harassment triggers the aggressive responses that characterise it [2]. This has led to the population fragmenting into subpopulations, traditionally called ‘encastes’, with different phenotypic characteristics and different gene flow levels among them [3].

Several studies have determined that aggressiveness (the animal’s ability to confront or to try to escape) and ferocity (the amount of force it uses to attack with its whole body and its resistance to pain) have strong genetic [4] and environmental bases [5]. However, other studies have found that even hard-tempered cattle (including fighting bulls) and other species eventually habituate to new environmental conditions and reduce their behavioural reactivity [6]. The mechanisms that underly the aggressive behaviour have been studied in numerous species, and a number of neurophysiological hormones and neurotransmitters are involved in the development of aggressive behaviour [7]. A well-established molecule implicated in the neurobiological mechanism of aggression is serotonin. Serotonin (5-hydroxytryptamine, or 5-HT) is a neurotransmitter monoamine and a hormone synthesised mainly in serotonergic neurons in the central nervous system, in the pineal gland and in the enterochromaffin cells (Kulchitsky cells) in the gastrointestinal tract. The main source of serotonin is the platelets in the blood circulation [8].

Serotonin plays an important role as a neurotransmitter in aggression inhibition and endocrine regulation, along with appetite and body temperature control, mood, sleep, cardiovascular functions, muscle contraction, learning and memory [9]. Low serotonergic low activity has been associated with a history of aggressive behaviour [7,10,11,12,13], both in primates and in humans with impulsive-type aggression and not with controlled or predatory aggression [11,14,15,16].

The relationship between low serotonergic activity and impulsive aggression has also been tested in different animal models such as mice, rats, and dogs. Several studies indicate that, in general, serotonin has an inhibitory effect on the brain [11,17] and is deeply involved in the regulation of emotion and behaviour, including the inhibition of aggression [18]. Therefore, it can be argued that serotonergic circuits are determinants in the control of aggression in humans and other animals [19,20].

Dopamine, in contrast, is a hormone and neurotransmitter that has several functions in the central nervous system, including behaviour and cognition control, motor activity, motivation and reward, milk production regulation, sleep, mood, attention and learning [21]. These monoamines are recognised for their capacity to modulate aggressive behaviour. Whilst serotonergic neurons that execute such influence have been identified, less is understood about the specific role of dopaminergic neurons in controlling aggression [22].

Pharmacologically induced dopamine increases are associated with increased aggressive behaviour under some conditions. Low to moderate doses of amphetamine or apomorphine can heighten the aggression levels of isolated mice or rats, whereas higher doses of amphetamine increase the defensive responses of rats [23]. The dysregulation of dopamine mechanisms in response to environmental stimuli leads to the perception of these stimuli as threatening. This decrease could be involved in the occurrence of affective or defensive aggression responses [24]. However, increased dopaminergic activity of limbic regions facilitates impulsive-type aggression in rats [25].

In addition to these neurotransmitters, steroid hormones such as testosterone are also involved in aggressive behaviour. Testosterone is a steroid hormone that is produced in the adrenal cortex and in the male and female gonads. In vertebrates, testosterone is involved in the modulation of sexual and aggressive behaviour. The relationship between elevated testosterone activity and aggressive–competitive–dominance behaviour is well established in all mammals. This type of response is termed hormonal or testosterone-dependent aggression. However, testosterone does not directly change behaviour, but increases the probability that a behavioural response will occur in the presence of specific stimuli [26]. Specifically, some authors indicate that aggressive and dominance behaviour in non-human primates and humans is less dependent of androgen mechanisms and that neurochemical control must be the main mechanism of aggressive behaviour [27].

In the brain, androgen and oestrogen receptors are mainly located in the limbic system [26], although they can also be found diffusely distributed throughout the brain. Testosterone implantation in the hypothalamus and caudate nucleus of castrated male rats can restore aggressive behaviour eliminated by castration [28].

Intraspecific social aggression (territorial, inter-male, rank-related) is a form of hormonal aggression in non-human mammals [29]. This form of competitive hostility is also observed in females, especially during lactation and offspring care (maternal aggression) [30]. Also, Peterson and Harmon-Jones (2012) associate the subjective experience of anger in humans with an increase in testosterone but not cortisol concentrations [31].

These factors are involved in the aggressive behaviour of several animal species. By its nature, the fighting bull is defined as an aggressive animal [2]. During the fight, the animal is exposed to numerous stimuli that promote changes in the animal’s behaviour. However, the mechanisms underlying these behavioural changes in bulls during fights have been little studied.

The aim of this study is therefore to determine the changes in the neuroendocrine variables serotonin, dopamine and testosterone during fights and their relation with aggressive behaviour. In addition, this study investigates whether these neuroendocrine variables are constant throughout the life of these animals and related to the development of aggressive behaviour, with the aim to use them as markers for the selection of fighting bulls.

## 2. Materials and Methods

### 2.1. Animals

A total of 195 male lidia bulls (*Bos taurus* L.) of different ages were used for this study. The animals in this study belonged to the following herds, all of which are registered in the Union de Criadores del Toro de Lidia (UCTL) of Spain: Jandilla, Fuente Ymbro, Juan Pedro Domecq, Núñez del Cuvillo, Cebada Gago, Victorino Martín, Adolfo Martin, Fermín Bohórquez, Urcola, Barcial and Francisco Galache. The animals studied belonged to six different “encastes” (all of them belonging to the lidia breed): Domecq (DQ), n = 37; Núñez (UN), n = 33; Albaserrada (AL), n = 32; Murube (MU), n = 34; Urcola (UR), n = 31 and Vega-Villar (VV), n = 28. An encaste is defined as follows: a strain, variety or closed population of animals of a breed which has been created based on reproductive genetic isolation for a minimum of five generations (RD 2129/2008, of 26-XII.).

Three groups of animals were established in this study:

Lidia calves: 180 calves between 6 and 8 months of age that were sampled at the time of shoeing.

Control: 15 bulls destined for bullfights that were not used in bullfights and were slaughtered in the slaughterhouse of the arena.

Lidia bulls: Of the 180 calves sampled at the time of shoeing, only 135 of them reached the age of 4 to 5 years old, since 45 animals did not reach the age of 4 to 5 years old for reasons unrelated to this study.

All procedures involving animals were reviewed and approved by the Animal Research and Ethics Committee of the Complutense University of Madrid (Reference number: UCM 0016/005).

### 2.2. Bullfighting

Bullfighting consists of a fight between a human (bullfighter) and an animal (bull) that ends with the death of the animal. It is governed by rules established by the Regulations of Bullfighting Shows [32]. Usually, the bull arrives at the arena and, hours before the start of the bullfight, it is isolated. When the bullfight begins, the bull is released into the arena, where it is received with a cape by the bullfighter. Afterwards, the bullfight is divided into three thirds:Third of sticks: the animal receives one or two pricks in the front area of the back.Third of skewers: three pairs of skewers are placed in the front area of the animal’s back.Third of crutch: The bullfighter passes the crutch to the bull and the fight ends with the death of the animal.

### 2.3. Blood Sample Collection, Processing and Hormonal Analysis

All animals were sampled at two different times: at the farrier stage (calves) and after the bullfight (bulls). Control bulls were sampled at 5 years of age.

In the calf groups, the animals were separated from their mothers 15 h prior to sample collection and kept in pens together until the time of marking. At that time, they were isolated and placed in a dressing box where they remained for 2 to 3 min. During this time, the animal was shod, and blood was obtained from the caudal vein.

Blood samples from controls and ordinary fighting bulls were obtained from the jugular vein immediately after the bullfight in the slaughterhouse of the arena. All blood samples were kept refrigerated at 4 °C from collection to processing. To obtain the serum samples, blood samples were centrifuged at 1200× *g*, 4 °C, for 20 min, and the serum was separated and stored frozen at −20 °C before being assayed. The serotonin, dopamine and testosterone concentrations were determined in serum samples using commercial kits (Demeditec Diagnostics GmbH, Kiel, Germany) following the manufacturer’s instructions. These are competition-type ELISA kits, where the competition is based on competition between the hormone in the sample and the hormone fixed to the solid phase, to bind the antibody. All commercial kits used were validated for bovine species, and the hormone concentrations are expressed in ng/mL. The detection limits were serotonin (6.2 ng/mL); dopamine (49 pg/mL); and testosterone (83 pg/mL). The intra- and inter-assays were less than 20% in all hormones analysed. The recovery rates were as follows: serotonin, 96%; dopamine, 89%; and testosterone, 93%.

### 2.4. Direct Observational Method

To evaluate the aggressive behaviour of bulls during a fight, a direct observational method was used by describing, through the observer’s perception, the bull’s stimuli and behavioural actions related to aggressiveness. These observations were recorded in a simple template expressly designed for this purpose that is detailed in the Appendix A.

The template was divided into three main parts. The observers scored these sections in a range from 1 to 5, with 1 being the partial score of an animal that did not present any of the aggressive actions and 5 the partial score of an animal that did develop the four aggressive actions specified in each part. Thus, for each aggressive action developed by a bull, one point would be added to each partial score. The average of the three partial scores, corresponding to the three parts of the bullfight, was calculated, which resulted in a preliminary score.

This allowed us to determine an aggressiveness score for their behaviour in a more objective way since this template only recorded the aggressive actions of the animal:Very slightly aggressive;Not very aggressive;Combative;Aggressive;Very aggressive.

### 2.5. Statistical Analysis

Graph Pad Prism version 6 (GraphPad Software, San Diego, CA, USA) was used for statistical analysis of the results.

To study whether there were differences among the different groups (bulls, calves and controls) with respect to the variables studied (serum concentrations of serotonin, dopamine and testosterone), an analysis of variance (ANOVA) was used. The distributions of values differed significantly between two groups at a *p*-value < 0.05. This test was also used to compare the variables cited in the different encastes.

The parameters studied were correlated with the grade of aggressive behaviour assigned to the animal after its fight. Pearson’s correlation test, a parametric test, was used in the case of the bull group. In the case of the calf group, a Spearman´s correlation test, a non-parametric test, was employed due to the smaller amount of data in this group.

Receiver operating characteristic (ROC) curve analysis was performed to determine if serotonin, dopamine and testosterone were parameters that were indicators of the behaviour developed during the bullfight. This type of analysis also allowed us to choose cut-off points that would facilitate decision making when classifying the behaviour of an animal based on the concentration of a given physiological variable.

For the comparison of the populations of expected and observed aggressive behaviour scores, a chi-square (χ2) statistical test was performed. Populations of values were determined to differ between the two groups when *p* < 0.05.

## 3. Results

### 3.1. Distribution of Aggressive Behaviour Scores in Bull Populations

After the application of the direct observational method, an aggressive behaviour score was obtained for each animal studied during the fights (n = 135). The percentages corresponding to each grade are detailed in Table 1; the most frequent grades were 3 and 4, and the least frequent ones were 2.5, 4.5 and 5.

The distribution of aggressive behaviour scores by breed was also analysed, revealing significant differences in aggressiveness scores among the different encastes (Figure 1). The cattle breeds with the lower average aggressive behaviour scores were Domecq, Murube and Núñez, whereas those with the highest aggressive behaviour scores were Vega-Villar, Urcola and Albaserrada.

#### Serum Serotonin Concentrations

No significant differences were found in serum serotonin concentration among the groups, although the controls had slightly lower serum serotonin concentrations (541.1 ± 4.9 ng/mL), similar to those observed in the calf group (546.7 ± 85.6 ng/mL) (Figure 2A). This indicates that serotonin concentrations are constant throughout the animal’s life, since there were no differences between the mean serotonin values at 6–8 months (calf group) of age and at 4–5 years of age (control group).

The serum serotonin concentrations of animals belonging to the different encastes differed significantly (Figure 2B). The lowest serotonin values were found in the animals belonging to the Urcola, Contreras and Albaserrada encastes. The encastes with the highest serum serotonin concentrations were Murube, Núñez and Domecq.

Regarding dopamine levels, no significant differences were found among the different groups studied (Figure 3A). However, the control group showed serum dopamine concentrations (10.78 ± 0.46 ng/mL) similar to those observed in the calf group (11.05 ± 0.50 ng/mL). This indicates that the dopamine concentration is constant throughout the animal’s life since there were no differences between the mean dopamine values at 6–8 months of age and at 4–5 years of age.

Also, the bull group showed higher serum dopamine concentrations (12.07 ± 0.50 ng/mL) compared to the control group, although this difference was not significant.

Comparison among the encastes revealed that the lowest dopamine values were found in the animals belonging to the Urcola, Vega-villar and Albaserrada encastes. The encastes with the highest serum dopamine concentrations were Murube, Núñez and Domecq. We observed statistically significant differences in the dopamine concentrations among breeds (Figure 3B).

### 3.2. Serum Testosterone Concentrations

We observed significant differences among the three groups studied (Figure 4A). A significant increase in testosterone concentrations was found in bulls (22.70 ± 0.52 ng/mL) and calves (9.74 ± 0.28 ng/mL) compared to the control group (4.22 ± 0.73 ng/mL). Bulls showed significantly higher testosterone levels than calves. However, the testosterone levels of the different encastes studied did not differ significantly (Figure 4B).

### 3.3. Correlation between Hormonal Concentrations Measured in Fighting Bulls and Aggressiveness Scores during Their Fights

A negative correlation was obtained between serum serotonin concentration and aggressiveness score, with a Pearson’s coefficient of −0.78 (*p* < 0.001; Figure 5A). Also, a stronger negative correlation was found in the case of dopamine concentrations, with a Pearson’s coefficient of −0.91 (*p* < 0.001; Figure 5B). These results suggest that the higher the serotonin and dopamine concentrations, the lower the aggressiveness score during the fight.

Nevertheless, a positive correlation was obtained between serum testosterone concentrations and the aggressiveness score, with a Pearson’s coefficient of 0.3675 (*p* < 0.01; Figure 5C). This indicates that there is a tendency that the higher the testosterone level, the higher the aggressiveness; however, the value dispersion was high.

Based on these correlations, a linear regression equation was obtained, which allowed us to estimate the expected aggressive behaviour score for an animal by interpolating its serum hormone concentration. For the different hormones, the linear regression equations were as follows:

Serotonin: y= −0.002262 x + 3.851.

Dopamine: y= −0.003145 x + 3.65.

Testosterone: y= 0.02814 x + 1.619.

### 3.4. Determination of the Hormonal Threshold Value in Fighting Bulls to Differentiate Subgroups of Animals with Different Behaviours

Given the correlations observed between hormonal concentrations and the grade of aggressive behaviour assigned to each bull, we wanted to determine whether serotonin, dopamine and testosterone concentrations are valid parameters that allow us to separate the population studied into subpopulations of animals showing different degrees of aggressiveness during fights.

Two possible classifications were considered (Figure 6). The first possible classification would be to separate the bulls into two groups (aggressive and non-aggressive), according to the score regarding their aggressive behaviour during their fight:-Aggressive: Animals that scored between 3 (inclusive) and 5. These are animals that clearly manifested aggressive actions during all parts of the fight.-Non-aggressive: Animals that scored between 1 and 3. These are the animals excluded from the previous group.

Another possible classification would be to separate the bulls by the lack of manifestation of aggressive behaviour during fights. In this case, the animals would be separated into combative and non-combative groups:-Combative: Animals with scores between 2.5 (inclusive) and 5. These animals showed behaviour of combativity and zeal, clearly showing aggressive behaviour or not.-Non-combative: Animals with scores between 1 and 2.5. These are the animals excluded from the previous group.

When the animals were sorted into the two different groups, they were further divided considering their serotonin and dopamine concentrations (Figure 7A,B). Non-combative and non-aggressive bulls presented significantly higher levels of dopamine and serotonin compared to combative and aggressive bulls.

Due to the significant differences found in the two possible classifications for both hormones, we performed ROC curve analysis for each hormone to determine which classification was the most appropriate for each hormone.

After performing the ROC curve analysis for serotonin concentrations, the animals that were classified as combative and non-combative showed an area under the curve of 0.9478 (*p* < 0.005). When the animals were classified into aggressive and non-aggressive groups, the area under the curve was 0.8878 (*p* < 0.005; Figure 8A).

In the case of the ROC curve analysis for dopamine concentrations, the area under the curve was 0.9216 (*p* < 0.001) when the animals were classified as non-combative and combative, and 0.9034 (*p* < 0.001) when the animals were classified as aggressive and non-aggressive (Figure 8B).

According to these results, it was decided to use the classification of combative/non-combative for both hormones as the most appropriate one since it presented higher areas under the ROC curve. This classification led us to determine the cut-off point in serotonin and dopamine concentrations that would allow us to classify the animals according to their behaviour.

The cut-off points chosen for serotonin and dopamine concentrations and their specificity and sensitivity values are summarised in Table 2. For serotonin, a cut-off point of 708.5 ng/mL was chosen, with a specificity of 90.63% and a sensitivity of 80.49%. In the case of dopamine, a cut-off point of 12.24 ng/mL was selected, with a specificity of 91.23% and a sensitivity of 82.50%. These results indicate that those animals with serum serotonin and dopamine concentrations below 708.5 and 12.24 ng/mL, respectively, can be classified as combative. Those animals with serum serotonin and dopamine concentrations greater than 708.5 and 12.24 ng/mL, respectively, were classified as non-combative animals.

The resulting specificity percentages (90.63% for serotonin and 91.23% for dopamine) indicated that those percentages of animals that were classified as non-combative were correctly classified, and the remaining ones (9.37% for serotonin and 8.67% for dopamine) were combative animals wrongly classified as non-combative ones.

Similar to the specificity percentages, the sensitivity values indicate that 80.49% and 82.50% of animals, for serotonin and dopamine, respectively, were detected with the chosen cut-off point, whereas the remaining ones (19.51% for serotonin and 17.5% for dopamine) were non-combative animals that were not identified correctly.

Regarding the testosterone concentrations, classification with respect to the aggressiveness scores of the bulls during the fights was carried out as in the case of the serotonin and dopamine concentrations (Figure 6).

When the animals were partitioned into the two different classes, the distribution of animals considering testosterone concentrations was as follows. In both classes, the two populations overlap (Figure 9A), resulting in a non-significant *t*-test analysis in both classifications (non-combative vs. combative, *p* = 0.1876; aggressive vs. non-aggressive, *p* = 0.0853). Due to this similarity between the different populations, it would be difficult to establish a threshold in testosterone concentration at which non-combative or aggressive behaviour can be discriminated.

After performing the analysis of ROC curves for testosterone, an area under the curve of 0.5794 (*p* = 0.29) was obtained for the animals classified as non-combative and combative, and an area under the curve of 0.6698 (*p* = 0.04) was calculated for the animals classified as aggressive and non-aggressive (Figure 10B).

Because the area under the curve obtained in both cases was significantly distant from 1, it was not considered appropriate to use testosterone as an indicator of aggressive behaviour; therefore, a threshold value was not determined from the ROC curve analysis.

### 3.5. Relationship between Serotonin, Dopamine and Testosterone Concentrations in Calves during Shoeing and Aggressive Behaviour during Their Subsequent Fights

The calves studied at the time of shoeing were followed up to evaluate their aggressive behaviour in the arena during their fights. Thus, we verified whether the concentrations of serotonin, dopamine and testosterone measured in the blood during the shoeing of these animals remained constant until the moment of their fights and whether they could be related to the manifestation of aggressive behaviour or not during their subsequent fights.

To estimate the expected aggressiveness score in calves, the linear regression equations and cut-off points previously calculated for each hormone in bulls were considered. Assuming these two results, the expected score was obtained by interpolating the calf serum hormone concentration in the linear regression equation.

Once the expected aggressive behaviour scores of calves were calculated, they were correlated with the hormonal values analysed in the serum at the time of shoeing (Figure 10). A clear negative correlation was obtained between serotonin and dopamine serum concentrations and the expected aggressiveness scores, with Spearman coefficients of −0.6653 and −0.8915, respectively. This indicates that the higher the serum serotonin and dopamine concentrations, the less aggressive the expected behaviour (*p* < 0.05).

Contrarily, a weak positive correlation was obtained between serum testosterone concentrations and the aggressiveness scores, with a Spearman coefficient of 0.3675, revealing a tendency towards more aggressive behaviour during the bullfight when higher serum testosterone concentrations were found (*p* = 0.2608).

Comparing the expected and observed aggressiveness scores with the results of the χ^2^ statistical analyses for serotonin and dopamine (Figure 11), values of *p* = 0.8035 and *p* = 8352, respectively, were found, indicating no significant differences between the expected and the observed scores. Thus, the value expected at the time of shoeing could be similar to that observed during the fight. Regarding the comparison of expected and observed scores for testosterone, an analysis could not be performed due to the lack of variability in the expected population.

## 4. Discussion

Although there are numerous scales that record the intensity of violent responses in humans [16,33], most of them refer to general aggression. Studies measuring different types of aggression in laboratory animals are also common [34]. However, behavioural assessment is complicated by not reflecting a unitary and clear motivational induction, since aggressive acts can express highly different feelings (anger, attack, defence, predation), triggered by different incitements (impulsivity or premeditation) and by complex factors (genetics or environments unfamiliar to each the subject) [9,35,36].

In addition to the described qualitative components, aggression has a complex physiological component in which different mechanisms and neurotransmitters play roles in promoting aggressive behaviours that vary among species [37]. Animal species such as the fighting bull have been described to be aggressive animals by nature [38], although this peculiarity of the fighting bull has been poorly studied. Therefore, this study aims to determine the physiological component that underlies the aggressiveness of fighting bulls in an observational manner, analysing their behaviour in fights and determining its correlation with the hormone levels related with aggressiveness in calves during their fights.

Bullfighting is, moreover, an activity with unique characteristics that requires different characterisation due to the aggressiveness of the fighting bull. In this regard, Gaudioso et al. (1993) designed a table to record the bravery of bulls during fights [39]. Since the design of this behaviour table, few studies have developed other models to measure bull behaviour during fights.

The bravery of a bull is a complex term to define objectively, and often, breeders do not agree on the characterisation of this bravery. Therefore, Gaudioso’s table is complex, with numerous parameters to be considered. The study carried out in fighting bulls by Calvo (2010) reflects this difficulty in defining the concept of bravery in an objective manner and shows the numerous definitions that different studies have presented for this concept [40]. One of the most widely used definitions of bravery for fighting bulls is the one that considers bravery as the bull’s capacity to fight until death [41].

Therefore, in this study, we designed an alternative observational table that allowed us to focus our attention in a simple way on the aggressive behaviour of bulls. In the observed distribution of marks obtained during fights by the bulls analysed, the majority of the bulls presented marks between 3.5 and 4, with the most frequent mark being 4 and with a mean of 2.55. The clearly non-aggressive animals (grade = 1) represented 5.96% of the total animals analysed, whereas the lowest percentage represented corresponded to the clearly aggressive bulls (grades between 4.5 and 5).

Since the observational measurements provided qualitative data of bull aggressiveness, we proceeded to assess whether these data correlated with the serum concentrations of hormones involved in aggressive behaviour, namely serotonin, dopamine and testosterone.

The involvement of the neurotransmitter serotonin in aggression inhibition, body temperature regulation, mood, sleep, sexuality and appetite has been demonstrated [42]. Serotonergic hypoactivity is related to antecedents of aggressive behaviour in humans and non-human primates [43,44,45], as well as in other animals [10,20]. Serotonin is considered to inhibit most forms of aggression, and predominantly that of impulsive (explosive and uncontrollable) characteristics rather than premeditated ones. Thus, serotonin deficiency, rather than increasing aggression itself, would hinder the control of impulsivity [9].

In this study, the serum serotonin concentrations in the different groups studied (calves, bulls and controls) did not vary significantly among them, although we did observe high intragroup variability.

The dopaminergic system is also involved in behavioural activation, motivated behaviour and reward processing [46,47,48]. It also plays an active role in modulating aggressive behaviours. In animal studies, hyperactivity in the dopamine system is associated with increased impulsive aggression [49,50]. In humans, the dopaminergic system has been linked to the recognition and experience of aggression. There is evidence that impulsive behaviour can be enhanced by an increase in dopaminergic function [51,52].

Based on the results obtained for serum dopamine concentrations, the serum dopamine levels in bulls during a fight were higher, but not significantly higher, than those of the control group, indicating that during a fight, the dopaminergic system is activated. Similarly, different studies on aggressive behaviours in rodents showed that elevated dopamine levels were observed before, during and after aggressive fights [53,54].

The serotonergic system has strong anatomical and functional interactions with the dopaminergic system [55]. More specifically, a reciprocal interaction between these two systems has been proposed [17]. Lunge- and withdrawal-related behaviours are thought to be determined by the balance between dopamine and serotonin activity [11], with dopamine increasing combative behaviours [56] and serotonin increasing more aggressive behaviours; the withdrawal of both leads to uncontrollable aggressive behaviours [17].

According to a previous study, there is an inverse association between dopamine and serotonin levels during aggression [57]. In this study, during fights, serotonin levels slightly decreased, while dopamine levels increased. This has also been observed in another study, which demonstrated that the serotonin levels in rats decreased by 80% from the basal level during and after fights, while the dopamine levels increased by 120% after fights [25].

Since the selection of fighting bulls in each herd is carried out according to the different criteria of each breeder [58], we wanted to study the tendency to show aggressive behaviours not only at a general population level but also separately considering the different breeds to which the herds studied belong.

There are encastes that traditionally show more manageable or more aggressive behaviours, depending on the cattle selection criteria (Rodríguez Montesinos, 2002). Although all breeders share the search for brave bulls, some put more effort into obtaining a more “docile” bull [2,41], whereas others emphasise obtaining a more “aggressive” bull [59].

When we analysed the serum serotonin and dopamine concentrations of the bulls according to their breed, significant differences in the concentrations of both hormones were found among the encastes. Those with the highest serum serotonin concentrations and the lowest dopamine concentrations were those that have traditionally been considered “more docile” encastes, such as Murube, Domecq and Núñez.

Overall, the differences in serotonin and dopamine concentrations found during the fights and among the encastes, and the stability of these hormonal levels throughout the lives of these animals, suggest that the serum serotonin and dopamine concentrations are good indicators of the level of aggressiveness manifested by bulls during fights. These results can be useful for breeders when choosing which encaste to breed.

To demonstrate the relationship between both hormones and aggressiveness, a correlation between serotonin and dopamine levels was performed, along with scoring the aggressiveness of the animals. We observed a strong negative correlation of both serotonin and dopamine concentrations with the obtained observational scores. Other studies reported similar results in other animal species [60,61].

Once this was established, the distribution of the aggressiveness scores was studied. The grouping of aggressiveness scores in the middle zone of the scale made it difficult to choose a cut-off point that would simplify the interpretation of the behavioural results and allow us to separate the animals into two groups: aggressive and non-aggressive.

Within the category of “aggressive” were included those animals that, during the course of a fight, clearly showed aggressive behaviour. This aggressiveness is easily perceived by the parties involved in the bullfight (the bullfighter, public and stockbreeder). The remaining bulls were included in the category of “non-aggressive” bulls.

However, with this first classification, the “non-aggressive” group contained animals that, although they did not show such manifested aggressive behaviour, showed certain combativeness and aggression. This type of aggressiveness is not particularly evident to the less trained observer, although it is a desirable selection trait for the breeder. Thus, bulls were divided into “combative” and “non-combative” groups.

This classification allows us to distinguish between “non-aggressive and non-combative” bulls, “combative and aggressive” bulls and “combative and non-aggressive” bulls. Non-combative bulls are those that do not show any combative or aggressive characteristics during a fight. In contrast, aggressive and combative bulls are those that are difficult to handle and complicated in a fight. Finally, there are combative and non-aggressive bulls, which are those most desired by bullfighters, breeders and the public because they show courage and bravery but are more manageable, allowing for a more aesthetic bullfight.

Once this classification of the bulls based on their aggressive behaviour during the fights was established, the determination of serotonin and dopamine levels as good parameters for discriminating combative and non-combative bulls, as well as aggressive and non-aggressive bulls, was performed. Both hormones allow the classification of bulls into aggressive or non-aggressive groups and into combative or non-combative groups. However, considering the differences between both classifications, for both hormones, serotonin and dopamine concentrations are more suitable for classifying bulls into combative and non-combative groups.

The interactions between the dopamine and serotonin systems provide a framework for understanding the mechanisms underlying aggression. Considering the functional regulation of serotonin over the dopamine system, impaired serotonergic function may result in hyperactivity of the dopamine system and the emergence of impulsive aggressive behaviours [11]. In addition, this interaction between dopamine and serotonin also has a genetic component, where dopamine activates impulsivity in rats that was increased by serotonin depletion or deletion of the serotonin-1B receptor gene [52,62]. Therefore, these interactions between dopamine and serotonin systems underlie aggressive behaviour that must be verified behaviourally. There are some studies in humans that correlate low levels of 5-HIAA (serotonin metabolite) and high levels of homovanillic acid (dopamine metabolite) with high scores in interpersonal and behavioural elements of psychopathy [63].

According to these results, that both dopamine and serotonin concentrations highly correlate with aggressiveness scores, both can be considered good parameters to classify bulls into combative or non-combative groups. However, it is difficult to classify an animal without knowing the threshold between being non-combative and combative. Therefore, cut-off values for serotonin and dopamine were calculated to enable a good classification. These values make it easy to classify bulls depending on serotonin and dopamine concentrations.

Predicting the behaviour of an animal based on a hormonal study and classifying it as combative or non-combative is highly complex. Therefore, observing the behaviour of the animal is necessary to establish this classification. In fact, it is easy to have animals previously classified as combative that, due to multiple factors, especially those related to the fight, are modified in their behaviour and therefore present a lower score of aggressiveness than what would correspond to them. However, this model is good when classifying non-combative bulls since it is more difficult to find an individual that, previously classified as non-aggressive due to its serotonin and dopamine concentrations, shows extremely aggressive behaviour during a fight.

Another hormone that is related to aggressive behaviour is testosterone. This androgen modulates sexual and aggressive behaviour, increasing the likelihood of aggressive behavioural responses in the presence of specific stimuli [64,65].

In the present study, testosterone levels significantly increased during fights in bulls compared to the control group, suggesting that testosterone is involved in the development of aggressive behaviour. In relation to this, studies with human participants also demonstrated a positive correlation between testosterone levels and the development of aggressive behaviours [66]. Although our results also found a positive correlation between testosterone levels and the aggressiveness score, this correlation was not statistically significant; therefore, our findings do not corroborate that testosterone levels are correlated with aggressiveness during fights in fighting bulls, in contrast to previous findings [64,65]. The increase found in testosterone levels during fights could be due to the physical exercise of the bulls and the intensity of the exercise, according to other reports [67].

When evaluating whether testosterone is a good parameter to predict aggressive behaviour, the testosterone concentrations were not useful in discriminating among bulls according to their behaviour. The area under the ROC curve was significantly far from 1, both in the case of bulls divided into aggressive and non-aggressive groups and those classified into combative and non-combative groups. Therefore, it was not possible to find a cut-off point for testosterone concentration that would allow us to classify bulls with good sensitivity and specificity values.

So far, no studies have investigated the physiological parameters involved in the aggressive behaviour of fighting bulls. Taking the results of this study together, serotonin and dopamine concentrations may be good parameters to detect animals with predictable non-combative behaviour, which would help breeders in the selection of fighting bulls. However, testosterone level does not seem to be an adequate parameter to detect more or less aggressive behaviour in bulls during their fights. Nevertheless, serum testosterone concentration can be used as an indicator of adaptation to physical exercise [67].

One goal of this study was the follow-up of the animals from the time of the shoeing of calves, when the animal is not yet sexually mature, until the time they reached sexual maturity and were bulls destined for the fight. This was useful to demonstrate that serotonin and dopamine concentrations remain stable throughout the animal’s life. In the present study, both serotonin and dopamine concentrations varied slightly between calves and control bulls, whereas other authors have reported a moderate decrease with age [68,69].

However, the serum testosterone concentration in the sexually mature control bull group was significantly lower than that in the calf group. As the testosterone concentration varies throughout an animal’s life, specifically at the stage of sexual maturation [70], the high levels obtained in the calf group may be due to the acute stress to which calves are subjected before and during shoeing [67].

Considering that the serotonin and dopamine concentrations did not present significant variations throughout the bulls’ lives and that both hormones were good aggressiveness parameters, the measurement of serotonin and dopamine levels in calves may allow us to calculate the expected aggressiveness score. Thus, using linear regression equations for serotonin and dopamine and interpolating the serum serotonin and dopamine concentrations measured in calves at the time of shoeing resulted in an expected aggressive behaviour score for each calf. When comparing the expected score with that recorded in the bullfighting of each animal, 4 or 5 years later, we found that there were no statistically significant differences between the expected and observed scores, confirming the validity of serum serotonin and dopamine concentrations as indicators of the tendency to manifest aggressive behaviour during a fight.

It is necessary to emphasise the importance that these results can have as an accessory tool to provide guidance in the selection of fighting bulls, for the selection of both the cattle used in fights and those used for reproduction. This approach is useful in the detection of those animals that, by presenting serum serotonin and dopamine concentrations above the established thresholds, have a tendency to manifest non-aggressive and non-combative behaviours.

## 5. Conclusions

Serum serotonin and dopamine concentrations are good indicators of a bull’s tendency to manifest aggressive behaviour during fights. Both hormones are markers that can be used in the selection of fighting bulls and are especially useful in the identification of those animals with a predisposition to be non-combative. However, serum testosterone concentration is not a good indicator of aggressiveness manifested by a bull during a fight.

The serum concentrations of serotonin and dopamine measured in calves during their shoeing allowed us to assign an expected behavioural score to each animal, which was confirmed 5 years later during the fights.

## Figures and Tables

**Figure 1 vetsci-11-00182-f001:**
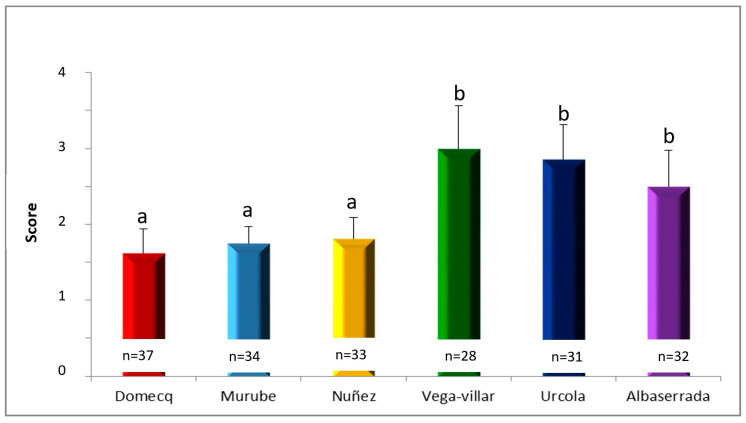
Distribution of aggressive behaviour scores in the different encastes studied, with different superscripts denoting significant differences (*p* < 0.05).

**Figure 2 vetsci-11-00182-f002:**
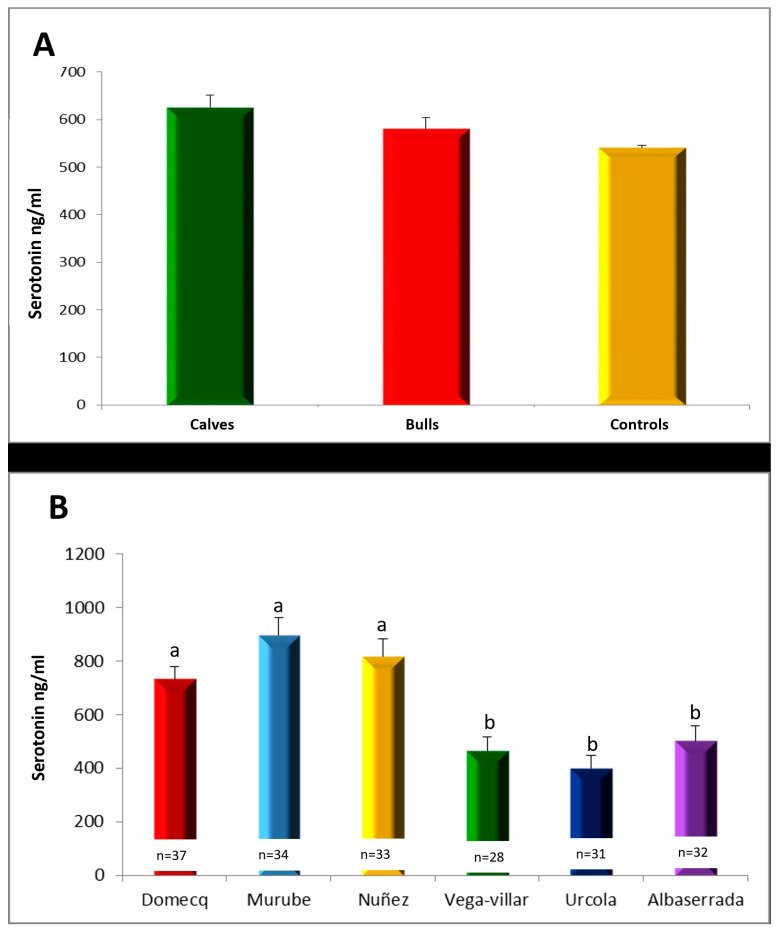
(**A**) Serum serotonin concentrations of the different groups studied. (**B**) Serum serotonin concentrations of the different encastes studied, with different superscripts denoting significant differences (*p* < 0.05).

**Figure 3 vetsci-11-00182-f003:**
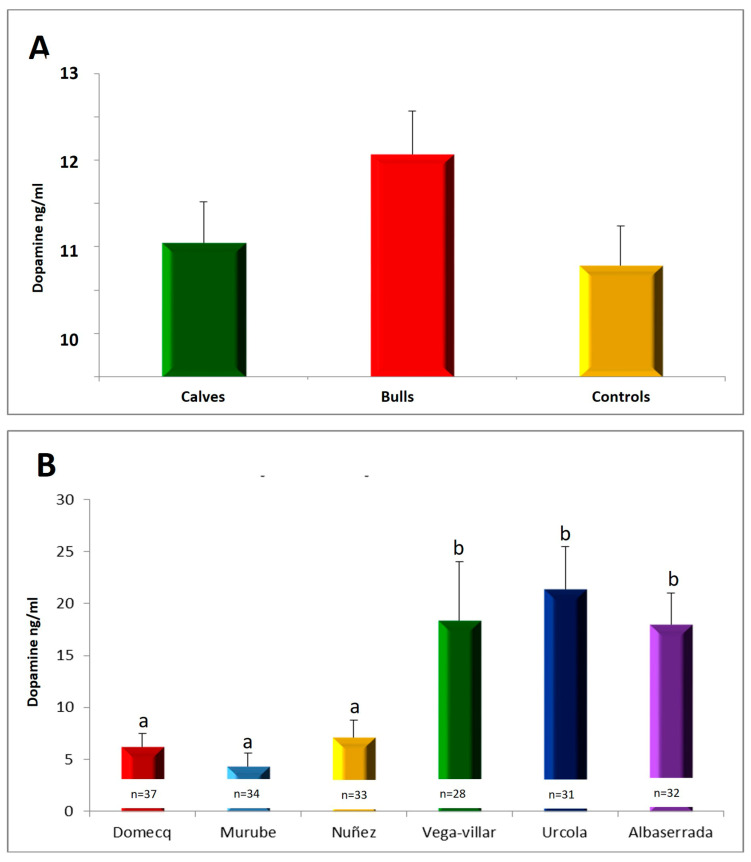
(**A**) Serum dopamine concentrations in the different groups studied. (**B**) Concentrations of dopamine in the serum of the different encastes studied, with different superscripts denoting significant differences (*p* < 0.05).

**Figure 4 vetsci-11-00182-f004:**
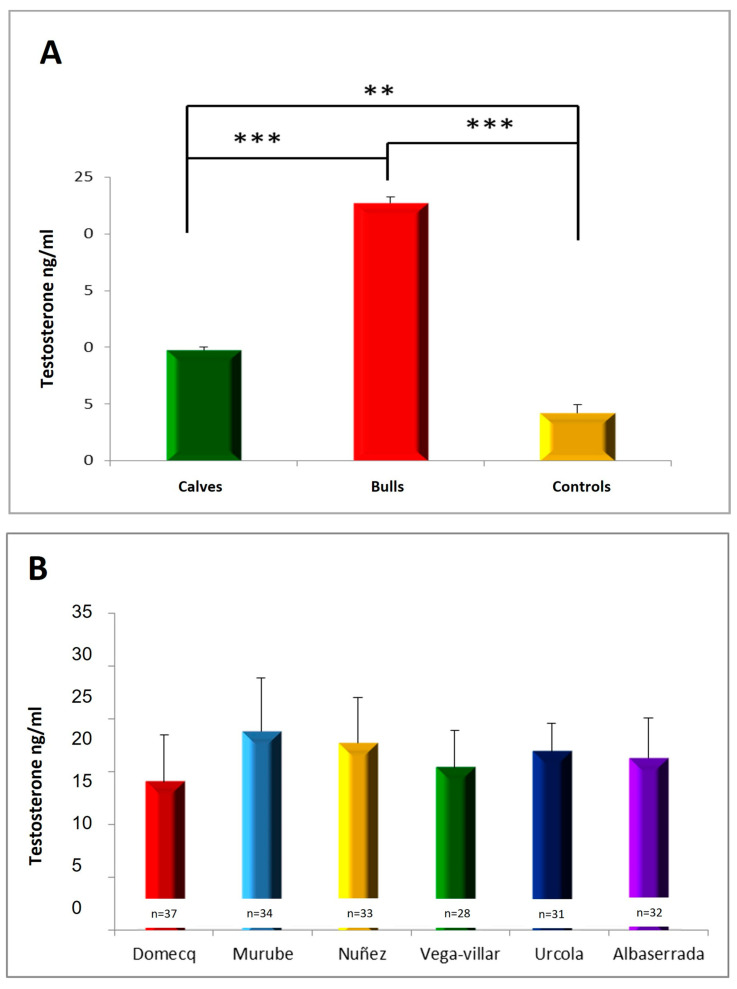
(**A**) Serum testosterone concentrations in the different groups studied. (**B**) Serum testosterone concentrations of the different encastes studied, with different superscripts denoting significant differences (*p* < 0.05). ** *p* < 0.01, *** *p* < 0.001.

**Figure 5 vetsci-11-00182-f005:**
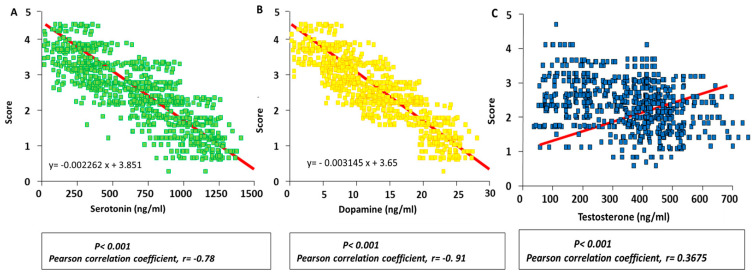
Correlations between serum serotonin (**A**), dopamine (**B**) and testosterone (**C**) concentrations analysed in bulls and the corresponding aggressiveness scores obtained during the fights.

**Figure 6 vetsci-11-00182-f006:**
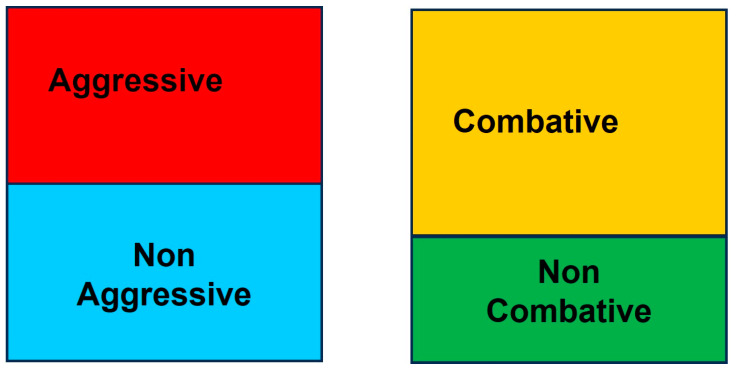
Classification of bulls according to their aggressive behaviour scores during fights.

**Figure 7 vetsci-11-00182-f007:**
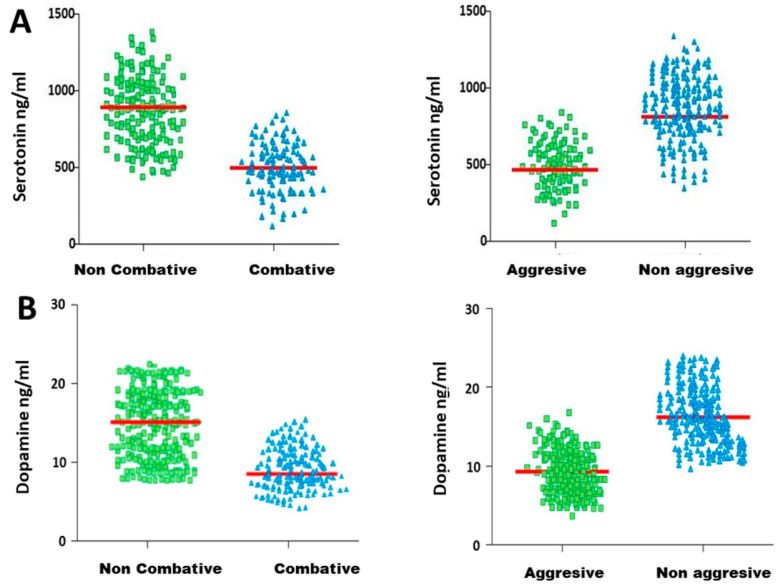
Serum serotonin (**A**) and dopamine (**B**) concentrations in the bulls divided according to their aggressive behaviour scores.

**Figure 8 vetsci-11-00182-f008:**
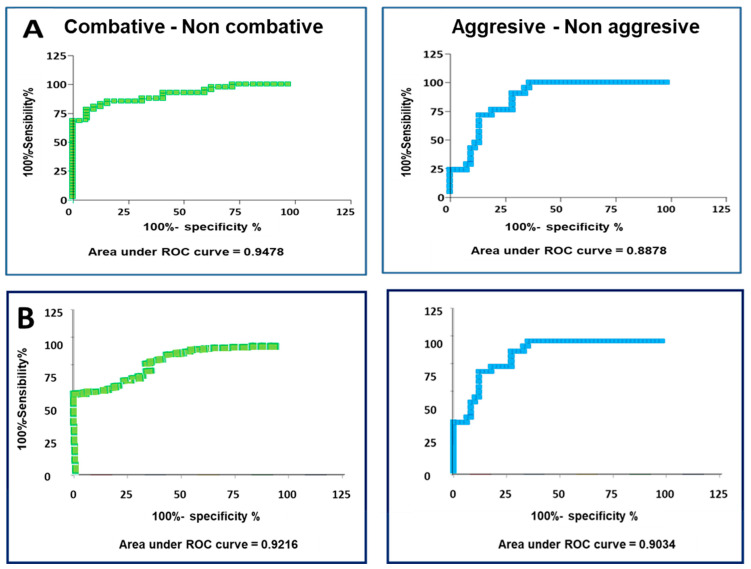
Receiver operating characteristic (ROC) curves for (**A**) serotonin–aggressive behaviour and (**B**) ROC dopamine–aggressive behaviour.

**Figure 9 vetsci-11-00182-f009:**
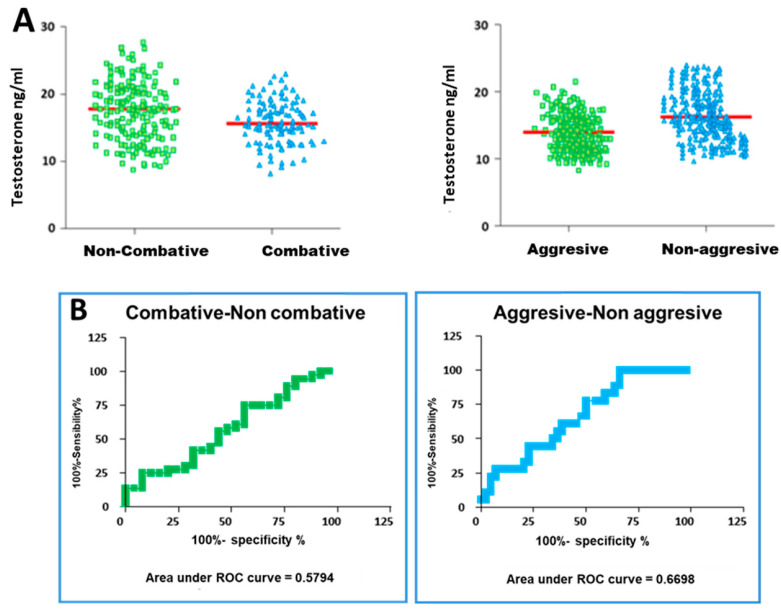
(**A**) Serum testosterone concentrations in the bulls classified according to their aggressive behaviour scores. (**B**) ROC curves for testosterone–aggressive behaviour.

**Figure 10 vetsci-11-00182-f010:**
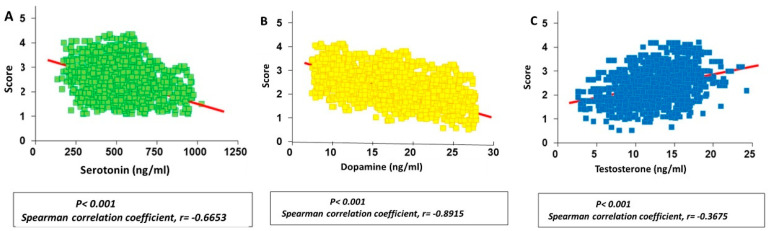
Correlation between serum serotonin (**A**), dopamine (**B**) and testosterone (**C**) concentrations analysed in calves and the corresponding previously calculated expected aggressiveness scores.

**Figure 11 vetsci-11-00182-f011:**
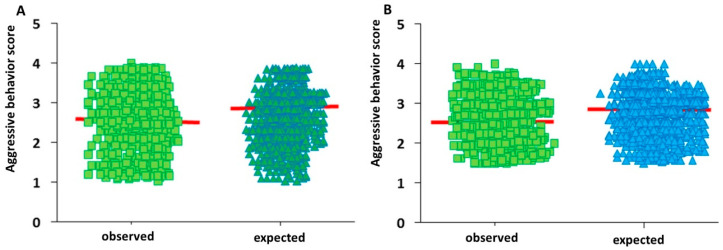
Expected and observed aggressiveness scores in calves as a function of serum serotonin (**A**) and dopamine (**B**) concentrations.

**Table 1 vetsci-11-00182-t001:** Distribution of aggressiveness scores of the total population studied during fights.

**Score**	1.0	1.5	2.0	2.5	3.0	3.5	4.0	4.5	5.0
**%**	5.96	5.96	17.89	3.15	25.61	11.22	26.31	3.15	0.35

**Table 2 vetsci-11-00182-t002:** Cut-off point values for serotonin and dopamine and the corresponding specificity and sensitivity percentages for each hormone.

	Cut-Off Point	% Specificity	% Sensitivity
Serotonin	708.5 ng/mL	90.63%	80.49%
Dopamine	12.24 ng/mL	91.23%	82.50%

## Data Availability

The data that support the findings of this study are available from the corresponding author upon reasonable request.

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
