# Peer review of "Addressing Combative Behaviour in Spanish Bulls by Measuring Hormonal Indicators"

_vetsci, 2024, doi:10.3390/vetsci11040182_

Round 1
Reviewer 1 Report
Comments and Suggestions for Authors
25 what does shoeing mean ?
31 Congratulations for introduction. The different aspects of agressive behaviour and its control are well described. Nevetheless, could you describe a litte bit more the Lidia breed namely also spanish bulls. All people are not very familiar with this breed.
119 Bos taurus L. L means ?
119 195 animals but you have used 180 calves, 15 control bulls and 135 bulls slaughtered in the arena after the fight
120 what does encastes mean ? Is it the right english word ?
152 what does shod mean ?
206 Threatening and not threatening
343 Fig 6 agressive and not aggresive
472 analyzed and not analysed
504 what does lunge mean ?
522 according to their breed : I believed that all the bulls had the same breed namely Lidia breed . Are you able to explain such differences between encastes ? Is it just a selection (by the breeder) effect ?
547 what does acometivity mean ?
Author Response
Reviewer 1
Comments and Suggestions for Authors
25 what does shoeing
mean ?
The word shoeing means the day on which calves are branded.
31 Congratulations for introduction. The different aspects of agressive behaviour and its control are well described. Nevetheless, could you describe a litte bit more the Lidia breed namely also spanish bulls. All people are not very familiar with this breed.
We have introduced the following paragraph
Lidia cattle are mainly used for breeding under extensive conditions in central and south-ern Spain, France, Portugal, and Latin America countries. This breed, also known as “toro bravo”, refers to the male specimens of a heterogeneous bovine population developed, se-lected and bred for use in different bullfighting spectacles. They come from the indigenous breeds of the Iberian Peninsula. They are characterized by atavistic instincts of defense and temperament, which are synthesized in the so-called "bravery", as well as physical attributes such as large horns forward and a powerful locomotor apparatus [2].
119 Bos taurus L. L means ?
The L stands for Linnaeus, and we have put it this way because the bibliography consulted said that this was the breed.
119 195 animals but you have used 180 calves, 15 control bulls and 135 bulls slaughtered in the arena after the fight
The total number of animals was 195, of which 15 were controls and the other 180 were sampled at the slaughterhouse, but due to circumstances beyond the control of the research, some animals did not get to be bullfighting, thus decreasing the number of bulls after the slaughterhouse to 135. We have clarified this division of the animals in the material and methods section of the manuscript.
120 what does encastes mean ? Is it the right english word ?
The word encaste means: strain, variety or closed population of animals of a breed, which has been created based on genetic reproductive isolation for a minimum of five generations. And in the fighting bull, there are several encastes from which all the bulls used in the bullfighting are derived. And if it is the right word because there is no other in the world of the fighting bull. Although if you suggest it, we can change it to strain.
152 what does shod mean ?
The word shod means to put the brand of fire on the calf.
206 Threatening and not threatening
We have changed the word in text.
343 Fig 6 agressive and not aggresive
We have changed the word in the figure.
472 analyzed and not analysed
We have changed the word in text.
504 what does lunge mean ?
The word lunge means: The action of the fighting bull when it attacks the bulge or the deception.
522 according to their breed : I believed that all the bulls had the same breed namely Lidia breed . Are you able to explain such differences between encastes ? Is it just a selection (by the breeder) effect ?
In fact, the breed is the same, the only difference is that with the different crosses that the breeders have been making, the different breedings have been obtained according to the different bullfighting parameters. These breedings are the origin of the current bulls that are fought worldwide.
547 what does acometivity mean ?
The word acometivity is defined as: Aggressiveness, inclination to attack, to attack verbally or physically. In the fighting bull with the bravery appears the acometividad, that is to say the attack with force against something, characteristic of the fighting bull that allows him to fight with cunning. It is the development of this characteristic that allows the fighting bull to be fought.

Reviewer 2 Report
Comments and Suggestions for Authors
The MS entitled “Addressing the combative...” by Illera analyzed the relationship of combative behaviour and serotonin, dopamine, and testosterone in Spanish bulls. Results revealed a strong
correlation between serotonin and dopamine levels, not testosterone and the aggressiveness scores in bulls during the fight,indicating that hormone determinations in calves may be great indicators for combative bulls and can reliably be used in the selection of fighting bulls.
General comments
The study study is interesting. However, there are some critical pitfalls for the publication of the MS. The authors used 195 claves and bulls from six different encastes. Also extra Lidia calves and bulls are used. Are they the same breeds? What’s the differences between them?
t -test is not appropriate for mutilple comparsons of different groups (bulls, calves and
controls) or for different encastes.
The writing of the MS needs to be improved. Some paragraph should be combined, and references should be cited,such as inL140-145, L196-200. In addition, L215-216 no need to show as this form in the MS. Too many in the present MS which need to be improved.
Specific comments
L147 all animals means what?L119: 195? or L125-130 :180+15+135?
L158 commercial kits, what detailed methods applied ?
L243 need a title for the table
L246-248 the text is no need to repeat within the figure. In addition,what does the lower panel of figure 1 mean? The differences among different groups?If yes ,pls mark them in the figure directly by using letters. Same as in figure 2, 3,4
L252 the serotonin level from different stage seems no difference, my question is what the breed is? The authors need to tell the sample number (n=?) and breeds. Actually, the serotonin level of different encastes is not the same of have significance.
L330-L346 any references for the classification of the aggressive and combative or non-aggressive and non-combative?
L370 Figure 8 what does it mean? How is it produced?
Comments on the Quality of English LanguageExtensive editing of English language required.
Author Response
Reviewer 2
Comments and Suggestions for Authors
The MS entitled “Addressing the combative...” by Illera analyzed the relationship of combative behaviour and serotonin, dopamine, and testosterone in Spanish bulls. Results revealed a strong correlation between serotonin and dopamine levels, not testosterone and the aggressiveness scores in bulls during the fight,indicating that hormone determinations in calves may be great indicators for combative bulls and can reliably be used in the selection of fighting bulls.
General comments
The study study is interesting. However, there are some critical pitfalls for the publication of the MS. The authors used 195 claves and bulls from six different encastes. Also extra Lidia calves and bulls are used. Are they the same breeds? What’s the differences between them?
Thank you very much for your comment. In all cases they are bulls and steers for bullfighting. The encaste come from crosses of different cattle ranches to obtain bulls with greater nobility and that are more apt for the fight, but they all continue being of the same breed. In some bibliographic data, the encaste are considered as strains, as it happens with other breeds that have strains, but they are of the same breed.
t -test is not appropriate for mutilple comparsons of different groups (bulls, calves and controls) or for different encastes.
Thank you very much for your appreciation, indeed we have made a mistake and when looking at the statistical data we have not used the t test for the different groups but we have used the ANOVA test.
We have changed the test in the manuscript.
The writing of the MS needs to be improved. Some paragraph should be combined, and references should be cited,such as inL140-145, L196-200. In addition, L215-216 no need to show as this form in the MS. Too many in the present MS which need to be improved.
Thank you for your comments. But the problem is that we have followed the guidelines of the Royal Decree that governs bullfighting shows in Spain. And we understand that it is complicated to write it well in English because the bullfighting vocabulary is generalized in Spanish. We have consulted official translators and they have told us that this is the English translation. And also we sent to review the English before sending it to the journal for consideration.
Specific comments
L147 all animals means what?L119: 195? or L125-130 :180+15+135?
We have changed the division into animal groups to better understand the number of animals used.
L158 commercial kits, what detailed methods applied ?
They are competitive ELISAs and we have introduced the following paragraph in the manuscript:
These are competition-type ELISA kits, the competition is based on the competition be-tween the hormone in the sample and the hormone fixed to the solid phase, to bind the antibody.
L243 need a title for the table
We have entered the title of the table in the manuscript.
L246-248 the text is no need to repeat within the figure. In addition,what does the lower panel of figure 1 mean? The differences among different groups?If yes ,pls mark them in the figure directly by using letters. Same as in figure 2, 3,4
Thank you very much for your comments. We have modified the graphs with the superscripts, the number of animals and we have changed the figure legend.
L252 the serotonin level from different stage seems no difference, my question is what the breed is? The authors need to tell the sample number (n=?) and breeds. Actually, the serotonin level of different encastes is not the same of have significance.
The breed is always the same, the number of animals is always the same in the different breedings as we have stated in the material and methods section. Being 28 the minimum number of animals used per group. But if you want we introduce in all the figures the n of the number of animals. And if serotonin and dopamine are of importance among the different breedings, since the differences found in the response of the bulls to the fight could be due to the differences in the levels of these hormones.
L330-L346 any references for the classification of the aggressive and combative or non-aggressive and non-combative?
There is no reference as it is a classification that we have made with our own results.
L370 Figure 8 what does it mean? How is it produced?
The reliability of the cut-off point of the hormones used, and whether that cut-off point corresponds to the aggressiveness or combativeness of the animal, compares the scores with the hormone levels studied. The definition of ROC curve is: The ROC curve is a statistical tool used to evaluate the discriminative ability of a dichotomous diagnostic test. These are curves in which the sensitivity is presented as a function of false positives (complementary to specificity) for different cut-off points. They are useful to choose the most appropriate cut-off point for a test, to know its overall performance and to compare the discriminative ability of 2 or more diagnostic tests.
We have used these ROC curves because the statistical team at the university told us to do so.
Comments on the Quality of English Language
Extensive editing of English language required.
Thank you for the appreciation. We have already sent the manuscript for professional English editing and we attached the following certificate.

Round 2
Reviewer 2 Report
Comments and Suggestions for Authors
L134: when were the 180 calves were slaughtered? As in L138, part of them at the age of 4-5 years.
L172: what’s the sensitivity and co-efficiency of the ELISA methods?
Page 4- page 7: L174 to L229 Direct observational method: this part should be simplified in the material and methods part by citing references or shown as supplemental data .
L260: breeds with the lowest: not lowest, but lower. Furthermore, there is no need to tell the detailed data as “Domecq (1.62), Murube(1.75) and Núñez (1.81), Vega-Villar (3.00), Urcola (2.86) and Albaserrada (2.75) ” in the text.THE SAME AS text related to FIGURE 2
L263: the markers for significant differences is not right. Are there any differences in the first three strains all marked with “a,b,c”? since they are all “a,b,c”, it means no significant difference. The scores of different groups maybe a (the left three) and b(the right three strains, more letters are no meaningful. THE SAME AS FIGURE 2, 3
Comments on the Quality of English LanguageMinor editing of English language required.
Author Response
Reviewer 2
L134: when were the 180 calves were slaughtered? As in L138, part of them at the age of 4-5 years.
Thak you for your suggestions. You were right, there was a mistake on the description of the Lidia calves group and also we have rewritten all that section as follows:
Lidia calves: 180 calves between 6 and 8 months of age that were sampled at the time of shoeing.
Control: 15 bulls destined for the bullfight that were not used in the bullfight and were slaughtered in the slaughterhouse of the arena.
Lidia bulls: of the 180 calves sampled at the time of shoeing, only 135 of them reached the age of 4 to 5 years old, since 45 animals did not reach the age of 4 to 5 years old for reasons unrelated to the study.
L172: what’s the sensitivity and co-efficiency of the ELISA methods?
We have introduced the validations parameters that you requested on the material and methods sections:
The detection limits were: serotonin (6.2 ng/ml); dopamine (49 pg/ml); and testosterone (83 pg/ml). The intra- and inter-assays were less than 20% in all hormones analyzed. The recovery rates were: serotonin 96%; dopamine 89% and testosterone 93%.
Page 4- page 7: L174 to L229 Direct observational method: this part should be simplified in the material and methods part by citing references or shown as supplemental data.
We have simplified all this section and introduced some details regarding the observational table as a supplementary file.
L260: breeds with the lowest: not lowest, but lower. Furthermore, there is no need to tell the detailed data as “Domecq (1.62), Murube(1.75) and Núñez (1.81), Vega-Villar (3.00), Urcola (2.86) and Albaserrada (2.75) ” in the text.THE SAME AS text related to FIGURE 2
Thank you, we have changed it in text. We also delated the data in the test regarding figure 1, and 2.
L263: the markers for significant differences is not right. Are there any differences in the first three strains all marked with “a,b,c”? since they are all “a,b,c”, it means no significant difference. The scores of different groups maybe a (the left three) and b(the right three strains, more letters are no meaningful. THE SAME AS FIGURE 2, 3
Thank you for your suggestion. We have changed the figures.
